# A Deep Temporal Neural Music Recommendation Model Utilizing Music and User Metadata

**Hai-Tao Zheng** [1,*], **Jin-Yuan Chen** [1], **Nan Liang** [1], **Arun Kumar Sangaiah** [2], **Yong Jiang** [1] **and Cong-Zhi Zhao** [3]

[1] Graduate School at Shenzhen, Tsinghua University, Shenzhen 518055, China; jy-chen13@mails.tsinghua.edu.cn (J.-Y.C.); liangn15@mails.tsinghua.edu.cn (N.L.); jiangy@sz.tsinghua.edu.cn (Y.J.)
[2] School of Computing Science and Engineering, VIT University, Vellore 632014, India; sarunkumar@vit.ac.in
[3] Giiso Information Technology Co., Ltd., Shenzhen 518055, China; zhaocz@giiso.com
[*] Correspondence: zheng.haitao@sz.tsinghua.edu.cn; Tel.: +86-180-3815-3089

**Abstract:** Deep learning shows its superiority in many domains such as computing vision, nature language processing, and speech recognition. In music recommendation, most deep learning-based methods focus on learning users' temporal preferences using their listening histories. The cold start problem is not addressed, however, and the music characteristics are not fully exploited by these methods. In addition, the music characteristics and the users' temporal preferences are not combined naturally, which cause the relatively low performance of music recommendation. To address these issues, we proposed a Deep Temporal Neural Music Recommendation model (DTNMR) based on music characteristics and the users' temporal preferences. We encoded the music metadata into one-hot vectors and utilized the Deep Neural Network to project the music vectors to low-dimensional space and obtain the music characteristics. In addition, Long Short-Term Memory (LSTM) neural networks are utilized to learn about users' long-term and short-term preferences from their listening histories. DTNMR alleviates the cold start problem in the item side using the music medadata and discovers new users' preferences immediately after they listen to music. The experimental results show DTNMR outperforms seven baseline methods in terms of recall, precision, f-measure, MAP, user coverage and AUC.

**Keywords:** semantic music recommendation; deep neural network; cold start problem

## 1. Introduction

With the development of mobile internet, enjoying online music by mobile phones is affordable for most users. Users may open their phones and listen to music anywhere, anytime. Many music recommendation systems use collaborative filtering (CF) methods [1–3]. CF methods, however, rely on explicit rating values to find similar users or items, while in music platforms, users listen to music but rarely rate the music. Although implicit feedback methods are alternative choices, these methods require careful parameter tuning [4,5]. All the methods cannot address the cold start problem. When new items or users appear, the corresponding model must be retrained to update the items' or users' profiles.

Recently, deep learning methods are incorporated into music recommendation systems to extract the users' temporal preferences and to solve the cold start problem. The music content based method learns latent factors of music to solve the item side cold start problem [6]. This method needs existing music representations which are commonly generated by a CF method. The performance is highly relevant to the chosen CF method. The user playlists based method represents music by pre-trained

word2vec [7] results and learn users' preferences based on their listening histories [8]. Therefore, only the songs that have appeared in playlists are trained for recommendation. None of them address the user side or item side cold start problem, and the music characteristics are not fully exploited. The music characteristics and the users' temporal preferences are therefore not combined naturally, causing the relatively low performance of music recommendation.

In order to solve these issues, we proposed a Deep Temporal Neural Music Recommendation model (DTNMR) based on music characteristics and users' temporal preferences. DTNMR utilizes Deep Neural Networks (DNN) to extract music characteristics and users' intrinsic preferences from their metadata. Additionally, Recurrent Neural Networks (RNN) are integrated to learn a users' long-term and short-term preferences from their listening histories. Since the music characteristics are extracted from music metadata, new and unpopular songs are recommended as same as other songs. User preferences are automatically updated when they listen to new songs, and the recommendation results varies accordingly. Therefore, DTNMR can recommend music for newly appeared users without retraining the model. As a consequence, DTNMR alleviates the cold start problem in item side using the music metadata and discovers a new users' preferences immediately after they listen to music by integrating DNNs and RNNs. Our contributions are listed as follows:

1. We propose a Deep Temporal Neural Music Recommendation model based on music characteristics and users' temporal preferences. DTNMR alleviates the cold start problem in item side and discovers new users' preferences immediately after they listen to music.
2. Based on music metadata and user profiles, DTNMR utilizes Deep Neural Networks to extract music characteristics and users' intrinsic preferences. The music characteristics of new songs can be extracted as same as popular ones.
3. Recurrent Neural Networks are used to extract users' dynamic preferences. The user preference varies when new songs are listened by the user. As soon as users listen to music, their preferences are updated and new recommendations are provided.
4. We conduct a series of experiments and show that our model performs the best among the seven baseline methods in terms of precision, recall, f-measure, MAP, user coverage and AUC.
5. We proved that our model can recommend songs for new users after they listened to a mount of songs without retraining.

The paper is organized as follows: The next section carries a review of the related work in the recommendation area. Section 3 elaborates on the Deep Temporal Neural Music Recommendation model. Section 4 shows the evaluation of our method. The last section presents the conclusion and the scope for future work.

## 2. Related Work

Collaborative filtering is a popular method for recommendation systems. The basic idea of collaborative filtering is recommending items according the users with similar interests [1] or recommending items similar with the items liked by the corresponding user [2]. These methods need to calculate similarities between each pairs of users or items and costs a great deal of computing resources. Consequently, matrix-based methods are proposed to solve the computing problem. These methods include Matrix Factorization [3–5], Probabilistic Latent Semantic Analysis [9], Bayesian Networks [10] and Random Walk [11,12]. Do and Cao [13] integrate metadata into coupled poisson factorization model to handle not only sparse but also popular ratings in very large-scaled data.

Content based recommendation systems utilize item contents to calculate the similarities between items and recommend similar items for users. Text categorization techniques are used to recommend movies in Mak et al. [14]. FOAFing the music [15] describes users interests based on their FOAF (friends of a friend) interests and habits. The interests are used to search artists and bands that the user is interested in Debnath et al. [16]. utilize a linear regression framework to determine the weights of different features. Bogdanov and Herrera [17] evaluate the usage of metadata information in content

based recommendation systems. Soleymani et al. [18] compute the similarities between music based on their Mellow, Unpretentious, Sophisticated, Intense and Contemporary attributes. These attributes are extracted from the auditory modulation features of music.

Deep learning based methods have been successfully adapted to multiple domains including computing vision, nature language processing and speech recognition. It has also been applied to recommendation systems in recent years. The first deep learning based recommendation model is proposed by Salakhutdinov et al. [19]. They utilized Restricted Boltzmann Machines (RBM) to predict the rating of movies. Van den Oord et al. [6] utilized Convolutional Neural Network (CNN) to extract hidden information from music. Wang et al. [20] proposed a collaborative deep learning model which combined stacked denoising autoencoders [21] and collaborative topic regression together. Consequently, Ying et al. [22] proposed that the pair-wised learning model collaborative deep ranking is better than collaborative deep learning for Top-K recommendations. Kim et al. [23] combined Probabilistic Matrix Factorization with Convolutional Neural Network because the rating information was extremely sparse. Auto-Encoder is also used to recommend items by Sedhain et al. [24], Zhang et al. [25], Zhuang et al. [26] and Chen et al. [27]. Matsumoto et al. [28] utilized social metadata to construct music-user network and perform link prediction on the obtained network to recommend music or video for users. Wang et al. [29] learned music embedding results based on users' historical listening sequences and the metadata of music.

Recurrent Neural Network (RNN) is also used to recommend items. Hidasi et al. [30] utilized RNN to predict users' future behaviors in session-based recommendation scenarios. Wang et al. [31] and Smirnova and Vasile [32] modeled users' dynamic preferences with RNNs. Devooght and Bersini [33] proposed that RNN is also useful in normal collaborative filtering datasets. Song et al. [34] combined deep neural networks and RNNs to recommend articles for users. De Boom et al. [8] learned users' preferences with RNN and recommended music based on the distance between user preference and music.

As collaborative filtering methods rely on the user rating information, the items' and users' characteristics are neglected. The cold start problem is another weak point of CF methods. Content-based recommendation systems calculate item similarities based on their contents or descriptions. It's hard to compute music similarities, however, due to the lack of descriptions and the fuzziness of the auditory modulation features. Most deep learning methods rely on the contents/tags of the items, and are not suitable for music recommendation. The deep learning methods designed for music recommendation neglect the music characteristics. And these deep learning music recommendation methods also suffer from the cold start problem on user side or item side. In this work, we recommend music more precisely by taking the music characteristics and user temporal preferences into consideration. Additionally, we also solve the cold start problem on both user side and item side by extracting music characteristics from music metadata.

## 3. Deep Temporal Neural Music Recommendation Model

In this section, we elaborate the Top-K recommender system, Deep Temporal Neural Music Recommendation model (DTNMR). In DTNMR, we combine linear features and embedded features, users' dynamic preferences and intrinsic preferences, temporal information and non-temporal information together to recommend songs for users precisely.

### 3.1. Overview

As shown in Figure 1, the DTNMR system consists of a user preference component, a music feature component and the final prediction component. The user preference component combines user dynamic preferences and user intrinsic preferences together. The structure of the user intrinsic component is similar to that of the music feature component. Both of the two components utilize Deep Neural Network (DNN) to extract hidden features from the inputs. The input of the music feature component is the metadata of songs, such as artist names, genres, languages etc. The input of user intrinsic component includes the user's personal information (i.e., age, gender, location) and all the

songs listened by the user. User dynamic preference is constructed by the user's behaviours over time. The long-term preference is the user's preference over a long time period, while short-term preference is the preference over a short time period. We utilize Recurrent Neural Network (RNN) to extract the user's dynamic preference from the user's listening histories. The input of each time slot is the song features extracted by the music feature component. Therefore, the user preference component describes users' preferences comprehensively and richly. Lastly, a neural network with softmax output is used to predict a score of current item for the corresponding user.

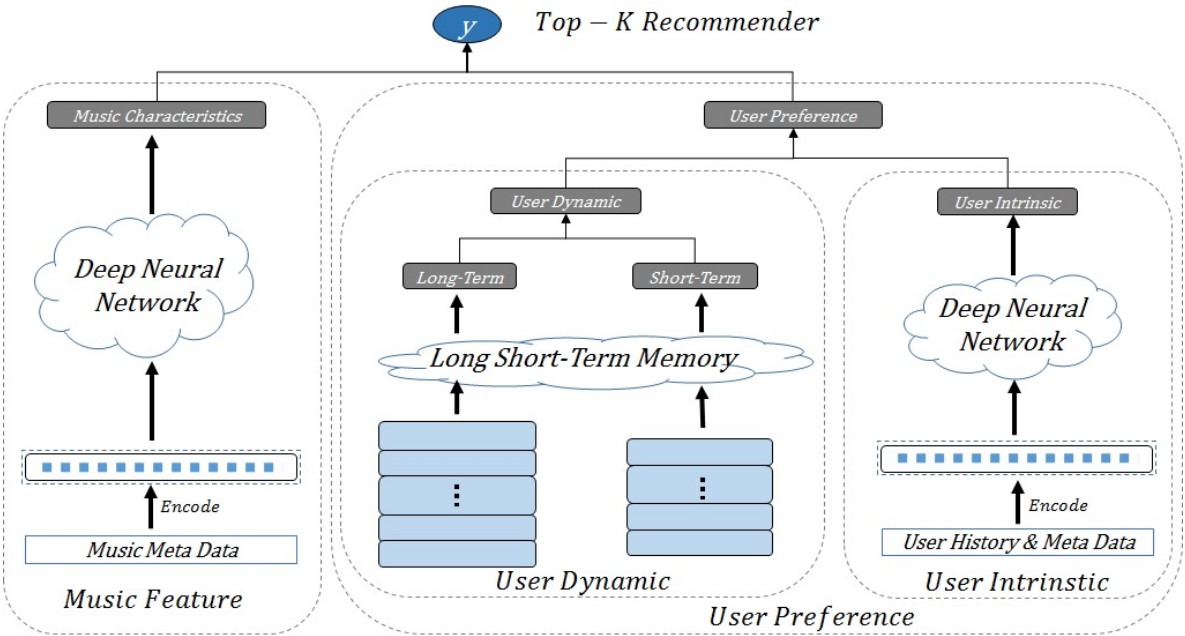

**Figure 1.** The Deep Temporal Neural Music Recommendation Model (DTNMR).

### 3.2. Music Feature Component

The music feature component extracts songs' hidden features based on song metadata. The meta fields used in the music feature component are listed in Table 1.

**Table 1.** The meta fields used in music feature component.

| Name | Type | Description |
| --- | --- | --- |
| Length | Linear | length of the song |
| Genre | One-Hot | genre of the song |
| Artist | One-Hot | artists of the song |
| Composer | One-Hot | composers of the song |
| Language | One-Hot | language of the song |

There are two field types in our model. Linear fields are the fields with continuous numeric values. The values of the linear fields are scaled into [0–1]. One-Hot fields are the fields with enumeration values. We encode the field values as one-hot vectors separately. We added the one-hot vectors together in case a song has multiple values, for example, a song may have several artists. The concatenation result of all the fields is the meta vector of the corresponding song. Assuming a song $s$'s length is $len(s)$, it has $n_g$ genres, $n_a$ artists, $n_c$ composers and $n_l$ languages, the meta vector of this song $M(s)$ is calculated as Equation (1).

$$M(s) = [len(s); (g_1(s) + ... + g_{n_g}(s)); (a_1(s) + ... + a_{n_a}(s)); (c_1(s) + ... + c_{n_c}(s)); (l_1(s) + ... + l_{n_l}(s))] \quad (1)$$

where $M(s)$ is the meta vector of $s$, $g_1$ is the one hot vector of the first genre, is the concatenation operation.

A Deep Neural Network (DNN) is used to extract hidden features from the meta vector $M(s)$. We calculate the output of layer $i$ as Equation (2).

$$l_i = ReLU(W_i \cdot l_{i-1} + b_i) \tag{2}$$

where $l_i$ is the output of the $i$th layer, $i = 1, ... N$, $l_0$ is the input of the DNN, $W_i$ is the weights of the neurons in $i$th layer and $b_i$ is the bias, $l_N$ is the output of the DNN and $ReLU$ is the activation function. In DTNMR, the input vector length of the DNN is about 20,000. The DNN consists of four layers; the first layer is the input layer, the second is a hidden layer with 512 nodes, the third layer contains 64 nodes, and the output layer contains 32 output nodes. The layer sizes are chosen empirically. The output item vector is denoted as $S(s) = l_N$.

*3.3. User Preference Component*

For the purpose of describing user preferences comprehensively and accurately, the user preference component takes user statistic results, personal information and user listening histories into consideration. The user statistic results and personal information are used to learn intrinsic preferences of the user. And user dynamic preference is learned from user listening histories over time. The user dynamic preference is further separated into short-term preference and long-term preference. The long-term preference learning procedure takes advantage of more user listening histories to learn preferences over longer time periods. The final user preference is the combination of the user static preference and the user dynamic preference as Equation (3).

$$U(u, t) = h(US(u), UD(u, t)) \tag{3}$$

where $U(u, t)$ is the user preference at time $t$, $US(u)$ is the user static preference, $UD(u, t)$ is the user dynamic preference at time $t$, and $h$ is an activation function, which can take one of the three forms:

$$h(W1, W2) = \begin{cases} W1 \odot W2 & \text{\textit{element wise multiplication}} \\ W1 + W2 & \text{\textit{element wise add}} \\ [W1; W2] & \text{\textit{concatenation}} \end{cases} \tag{4}$$

3.3.1. User Intrinsic Component

The User Intrinsic Component extracts a users' implicate intrinsic preferences through a Deep Neural Network (DNN). The inputs of DNN include the user's personal information and the statistic results of the user's behaviours. The fields used in user static preference model are listed in Table 2,

**Table 2.** The features used in user static preference model.

| Name | Type | Description |
| --- | --- | --- |
| Gender | One-Hot | The gender of the corresponding user |
| City | One-Hot | The city where the corresponding user lives |
| Age | One-Hot | The age group of the corresponding user |
| ListenCnt | Linear | The number of times the corresponding user listen to music |
| SongCnt | Linear | The number of songs the corresponding user listened to |
| SongEmbed | Added | The embedding features of the songs listened by the corresponding user |

Where SongEmbed is the One-Hot features in Table 1. We add the feature values of listened songs together as the user's SongEmbed feature.

$$SongEmbed = M'(s_u^1) + M'(s_u^2) + ... M'(s_u^n) \tag{5}$$

where $s_u^1, s_u^2, s_u^n$ are the songs listened by the user $u$, $M'(s_u^1)$ is the one-hot features in $M(s_u^1)$. We obtain $M'(s_u^1)$ by removing linear features from $M(s_u^1)$. The component structure is the same as that of music feature component, and the output is $US(u)$.

### 3.3.2. User Dynamic Component

The user dynamic component learns the user's long-term preference and short-term preference through recent activities of the user. The User dynamic component consists of two sub components, i.e., the user long-term component and the user short-term component. The two components share the same two layer RNN structure.

In addition, we find users may listen to songs in different ways (named as behavior type), i.e., through the playlist, search box or recommendation. The importance of listening histories with different behavior types are different. For example, a song in the user's playlist is more important than a song suggested by the system. In order to take behavior types into consideration, we encoded the users' behavior types as one-hot vectors. Further, the input vector for each time slot is the concatenation of the item-vector generated by the music feature component and the behavior type vector. Given a time span $\delta t$, the output of the RNN structure is denoted as Equation (6).

$$UD_{\delta t}(u, t) = RNN([S(s_u^{t_1}); B(u, t_1)], [S(s_u^{t_2}); B(u, t_2)], ..., [S(s_u^{t_n}); B(u, t_n)]) \tag{6}$$

where $u$ is the user, $t$ is the current time, $s_u^{t_1}, s_u^{t_2}, ..., s_u^{t_n}$ are the songs listened by user between $t - \delta t$ and $t$, $t_1$ is the first time $u$ listened to music after $t - \delta t$, $t_n$ is the last time, $B(u, t)$ is the behavior type at corresponding time and $RNN$ denotes the recurrent neural network.

In DTNMR, the users' long-term preferences and short-term preferences are added together and treated as user dynamic preferences. Therefore, the user dynamic preference is denoted as:

$$UD(u, t) = UD_{\delta t_{short}}(u, t) + UD_{\delta t_{long}}(u, t) \tag{7}$$

Since the user listening time is not provided in our dataset, we simulate $t_{short}$ by the latest 20 songs and $t_{long}$ is the latest 50 songs.

### 3.4. Rating and Training

Given the user preference $U(u, t)$ and the music characteristics $S(s)$, the rating value of the item for the user is the linear combination result of the input features:

$$y(u, s, t) = W_z[U(u, t); S(s)] + b_z \tag{8}$$

where $W_z$ and $b_z$ are the learned parameters. A user's final recommendation result is the Top-K items with the highest rating values.

The objective function is the softmax cross entropy loss, which is shown as follows:

$$L = -\sum_{u,t} \log P(u, s_u^t, t) \tag{9}$$

where $s_u^t$ is the song user $u$ listened at time $t$, $P(u, s_u^t, t)$ is the softmax probability of the song:

$$P(u, s_u^t, t) = \frac{e^{y(u, s_u^t, t)}}{\sum_{s \in \mathcal{S}} e^{y(u, s, t)}} \tag{10}$$

where $y(u, s, t)$ is the rating value of song $s$ for user $u$ at time $t$ and $\mathcal{S}$ is the song set. Ideally, the song set $\mathcal{S}$ should contains all the songs in the dataset. But in practise, it takes too much time to calculate all the rating results. Therefore, we utilized a small fragment of the dataset to approximate the loss. In our experiment, $\mathcal{S}$ is formed by the song $s_u^t$ and four randomly chosen un-listened songs from the dataset.

## 4. Experiments

### 4.1. Experimental Setup

The dataset used in our experiments was WSDM-KKBOX https://www.kaggle.com/c/kkbox-music-recommendation-challenge/data. KKBOX is the biggest online music streaming platform in Asia and the corresponding dataset was used in the 11th ACM International Conference on Web Search and Data Mining (WSDM 2018) challenge. The dataset contains the users' listening records during a certain time period. The records before a certain point are used as training data, and the records after this point are the test data. The time period and the separation point are chosen by WSDM. In addition, the metadata of songs and users is also provided. These metadata includes the song length, song genres, artist names, composer names, languages, user ages, user locations, user genders etc.

In the WSDM challenge, competitors are asked to predict the chances of a user listening to a song repetitively after the first observable listening event. In this work, the purpose of DTNMR was to recommend songs for users based on their listening histories, thus DTNMR helps the users find new interesting songs. In order to achieve this goal, we predicted the chances of a user listening to an un-listened song in the future and ranked all un-listened songs according to the predicted chances. Therefore, our task was different from WSDM in terms of input, output and also the training process. The differences of DTNMR and WSDM are elaborated in Table 3.

**Table 3.** The differences of DTNMR and WSDM challenge.

|  | DTNMR | WSDM |
|---|---|---|
| Input | The user listening histories | A user and a song the user previous listened to |
| Output | A list of ranked songs for suggestion | A probability indicating the user will listen to the song again |
| Train Input | The user, the listening histories and the current song | A user, a song the user listened and a boolean value indicating whether the user listened to the song again |

Thus, we could not compare our method with the methods in the WSDM challenge.

During pre-processing, the songs and users, which do not appear in both the training and test dataset, were removed. After that, we removed the records that appeared in the training dataset from the test dataset. This was because recommendation systems are used to help users find new interests, not to remind them of interests already known. At last, there are 34,403 users, 41,9781 songs, 7,377,304 training records and 2,236,323 testing records in the dataset. Table 4 shows the feature statistic results of the songs.

**Table 4.** Feature statistics of songs.

| Feature Name | Genre | Artist | Composer | Lyricist | Language |
|---|---|---|---|---|---|
| Feature Values | 172 | 49,411 | 92,663 | 43,526 | 10 |
| Feature num per song | 1.07 | 1.06 | 1 | 0.38 | 1 |

Seven baseline methods were used to evaluate the effectiveness of our method.

**MostPop**: The popularity-based method which recommend songs listened by most users to each user.

**SVD++**: Singular Value Decomposition based method which makes use of implicit feedback information [3].

**AoBPR**: Adaptive Oversampling Bayesian Personalized Ranking [5].

**WRMF**: A collaborative filtering method on datasets with implicit feedback [4].

**UserKNN**: Ranking items according the ratings of similar users [1].

**RBM**: The Restricted Boltzmann Machines (RBM) based deep learning method [19].

**LSTM**: The RNN based recommending method proposed by Devooght and Bersini [33].

As the UserKNN model needs to find Top-K similar users for each user-item group, it is too complex for a dataset with 30K users and 400K items. Therefore, we found the top 1000 similar users for each user, and rated songs according the top 50 users who listened to the corresponding song among these 1000 similar users. Since some of the methods were stochastic models, we ran these methods 10 times and calculated the standard deviation of these methods. We implemented the DTNMR and LSTM methods with Tensorflow 1.0.1. The other base lines were implemented by LibRec https://www.librec.net/, a GPL-licensed Java library which contains a suite of start-of-art recommendation algorithms.

The evaluation matrixes used in this work are listed as follows:

**P@10**: The precision of the top-ten results.
**R@10**: The recall of the top-ten results.
**F@10**: The f-measure of the top-ten results.
**MAP@10**: The mean average precision of the top-ten results.
**UCOV@10**: The user coverage of the top-ten results (e.g., the percentage of users received at least one correct item).
**AUC**: The area under the curve, equal to the probability that a randomly chosen interested song ranked higher than a randomly chosen uninterested song

We used the top-ten results because most recommendation systems recommend ten songs for the users each time.

## 4.2. Experiment Results

The first step of DTNMR is model training. The model training process is run in CPU mode (16 kernels), and takes less than six hours. In order to find the Top-K results for a user, we needed to compute the scores of each songs. Running the whole network was extremely slow since the song features are recalculated for each user. In practise, we utilized pre-calculated song features to recommend songs for users because song features are stable over time. In this case, it took about one second recommending for a user.

In order to determine the combination function $h$ in Equation (4), we randomly selected 90% of users as training data, and the other 10% users as validation data. After running each combination functions ten times, we did not find any significant difference between them. We used "element-wise add" function in all the experiments.

There are four components in DTNMR, the music feature component, the user long-term component, the user short-term component and the user intrinsic component. Among them, only the music feature is necessary in DTNMR; the other three components are not required by DTNMR. Additionally, in the user long-term component and the user short-term component, we could omit the behavior type of the listening histories. Therefore, our first experiment was set up to examine the influence of these components and behavior type. We conducted an experiment to evaluate the performances of DTNMR with different components.

As shown in Table 5, we proposed the results of five different combinations. The components U and L were used to extract users' dynamic preferences, while component L had more records than U, therefore, we chose component L to evaluate the contribution of the users' dynamic preference. The component B was the users' behavior types while listening to music, and it could not be applied to DTNMR independently. Therefore, there was only one experiment of component B. DTNMR-U was the worst among all the models. Because user preferences change over time, only considering user intrinsic preference was not enough. The performances of DTNMR-L, DTNMR-UL and DTNMR-ULS were similar. All of them took user temporal preferences into consideration with the help of LSTM. The short-term preference and the intrinsic preference could help DTNMR describe user preferences better, but they were less important than long-term preference. The DTNMR-ULSB performed the best because it considered the behavior type additionally. With the help of the user behavior type, DTNMR

was able to evaluate the different importance of user listening histories and models the user dynamic preferences in a better way. The Precision, Recall, F-Measure, MAP, User Coverage and AUC were 14.26%, 3.63%, 2.90%, 8.38%, 58.09% and 79.02% respectively.

**Table 5.** The influence of the four components in DTNMR (D denotes DTNMR for short, the component names after "-" mean the corresponding component is turned on, where S is user short-term preference, L is user long-term preference, U is user intrinsic preference and B is the behavior types).

| Method | P@10(%) | R@10(%) | F@10(%) | MAP@10(%) | UCOV@10(%) | AUC(%) |
|---|---|---|---|---|---|---|
| D-L | $13.2910 \pm 0.91$ | $3.4251 \pm 0.29$ | $2.7220 \pm 0.22$ | $7.6190 \pm 0.94$ | $56.4723 \pm 1.99$ | $78.2196 \pm 0.99$ |
| D-U | $13.0346 \pm 0.99$ | $3.2040 \pm 0.54$ | $2.5675 \pm 0.39$ | $7.5343 \pm 1.23$ | $54.9940 \pm 3.14$ | $77.4807 \pm 1.57$ |
| D-UL | $13.3159 \pm 1.20$ | $3.1545 \pm 0.42$ | $2.5464 \pm 0.31$ | $7.6950 \pm 0.88$ | $54.2578 \pm 2.84$ | $77.1127 \pm 1.42$ |
| D-ULS | $13.4966 \pm 0.91$ | $3.3300 \pm 0.40$ | $2.6683 \pm 0.28$ | $8.0431 \pm 1.14$ | $55.8836 \pm 2.21$ | $77.9254 \pm 1.11$ |
| D-ULSB | $\mathbf{14.2612 \pm 0.60}$ | $\mathbf{3.6349 \pm 0.21}$ | $\mathbf{2.8950 \pm 0.14}$ | $\mathbf{8.3833 \pm 0.53}$ | $\mathbf{58.0929 \pm 1.14}$ | $\mathbf{79.0296 \pm 0.57}$ |

Table 6 shows the performances of the seven baseline methods and DTNMR. DTNMR-ULSB was the best in terms of Precision, Recall, F-Measure, MAP and User Coverage. About 58% users were satisfied with at least one song in the top-ten recommendations.

**Table 6.** Comparison between DTNMR and the seven baseline methods.

| Method | P@10(%) | R@10(%) | F@10(%) | MAP@10(%) | UCOV@10(%) | AUC(%) |
|---|---|---|---|---|---|---|
| MostPop | 11.2547 | 2.4430 | 2.0073 | 7.1047 | 44.1653 | 72.0688 |
| SVD++ | 0.0051 | 0.0004 | 0.0004 | 0.0014 | $0.0509 \pm 0.01$ | $50.0254 \pm 0.01$ |
| AoBPR | $7.9151 \pm 0.41$ | $1.4077 \pm 0.09$ | $1.1950 \pm 0.07$ | $4.0941 \pm 0.30$ | $37.3732 \pm 1.33$ | $68.6738 \pm 0.66$ |
| WRMF | 10.6451 | 2.0716 | 1.7341 | 6.2792 | $42.1231 \pm 0.01$ | $71.0481 \pm 0.01$ |
| UserKNN | 13.5335 | 2.8104 | 2.3272 | 8.1964 | 52.0682 | 76.0176 |
| RBM | 0.0204 | 0.0015 | 0.0014 | 0.0066 | 0.2044 | 50.1021 |
| LSTM | $8.7646 \pm 0.77$ | $1.8996 \pm 0.14$ | $1.5611 \pm 0.12$ | $4.9397 \pm 0.56$ | $39.7728 \pm 2.06$ | $69.8735 \pm 1.03$ |
| D-ULSB | $\mathbf{14.2612 \pm 0.60}$ | $\mathbf{3.6349 \pm 0.21}$ | $\mathbf{2.8950 \pm 0.14}$ | $\mathbf{8.3833 \pm 0.53}$ | $\mathbf{58.0929 \pm 1.14}$ | $\mathbf{79.0296 \pm 0.57}$ |

SVD++ and RBM failed to recommend correct items for users. Their performances were no better than that of choosing songs randomly for users (0.02% in terms of P@10). This was because both of them are based on explicit item ratings. During our experiment, only user listening histories were provided.

The performances of implicit feedback based methods AoBPR and WRMF were much better than those of SVD++ and RBM. The precisions were 7.9% and 10.6% respectively. But these two methods could not handle the extremely sparse dataset, their performances are worse than our method.

The performance of LSTM was similar to AoBPR. LSTM model needed to train a neural network for each song. In music recommendation, however, most songs were listened to by a small number of users, and the neural network of these songs were not trained sufficiently. Therefore, the LSTM model also suffered from the sparse data problem.

MostPop recommended the same songs for all users. The precision of this method was 11.2% and the user coverage was 44.1%. We could infer that 44% users listened to at least one popular song, and some of them listened to multiple popular songs.

UserKNN was the second best method among all the compared methods. As UserKNN recommends items based on similar users, it does not suffer from the sparse data problem because we had enough users in the training set. Therefore, its performance was better than other baselines. The efficiency was the biggest problem of UserKNN. It needs to calculate the similarity between each user pairs and find the top users for each song. In our experiment, in which we found the top 1000 similar users for each user and recommended according to these users, the time taken for each user is about 10 s.

　　DTNMR recommends songs more precisely than other methods by considering music characteristics and users' temporal preferences. The music characteristics are extracted from their metadata by a deep neural network. The user preferences are extracted from the users listen histories with RNNs and DNNs. In addition, DTNMR overcomes the sparse data problem by using the metadata. Since metadata exists in all songs and users, DTNMR can describe songs and users more precisely. DTNMR outperformed UserKNN by 0.73%, 0.82%, 0.57%, 0.19%, 6.03% and 3.01% in terms of Precision, Recall, F-Measure, MAP, User Coverage and AUC. Compared with MostPop, these values are 2.71%, 1.19%, 0.89%, 1.28%, 14.07% and 6.96% respectively.

　　Finally, we conducted an experiment to examine the ability of recommending songs to new users and to demonstrate that DTNMR can learn new preferences for users. In this experiment, 10 percents of users were removed from the training set. After training the DTNMR with the remaining data, we utilized the trained model to recommend songs for the removed users. Each user's recommendation results were based on their listening histories in the original training set, and the evaluation result was based on the test set. The performance of recommending songs for these users of DTNMR was $13.6077 \pm 0.82$, $3.2026 \pm 0.41$, $2.5897 \pm 0.29$, $7.9849 \pm 0.72$ and $55.7587 \pm 2.24$ in terms of P@10, R@10, F-M@10, MAP@10, UCOV@10 respectively. This performance was similar with training DTNMR with the whole training set.

　　When DTNMR did not know any information about the users, it infered the behavior patterns from existing users and recommend new users accordingly. The long-term and short-term components described users dynamic preferences based on their recently actions. Therefore, users' dynamic preferences changed when they listen to new songs. Initially, there was no dynamic preferences for the new users, and after some actions, DTNMR described their dynamic preferences based on the new records. This characteristic was guaranteed by the LSTM mechanism. LSTM is developed for learning sequential information from sequence input. It's performance is verified in multiple domains such as language modeling, machine translation, speech recognition and image descriptions generation. In DTNMR, LSTM is used to learn users' preference from their recently actions, and the preference varied while the actions changed. The intrinsic preferences varied much slower than dynamic preferences since it recorded the users' entire histories. Table 7 shows the recommendation results of two users at different times.

**Table 7.** The recommendation results of user 1500 and user 13207 at different time.

| User 1500 | Recommendation Result |
|---|---|
| Time $t1$ | **246685** 170780 **366048** 415259 416067 419466 **212724** 152266 17662 |
| Time $t2$ | **246685** 170780 **366048** 415259 416067 419466 152266 **212724** 17662 |
| User 13207 | Recommendation Result |
| Time $t3$ | 366048 79938 416067 419466 119903 354948 **190396** 26854 308565 |
| Time $t4$ | 366048 79938 416067 419466 119903 354948 26854 **190396** 308565 |

　　In this table, bold songs are the correct recommendation. From this table, we found that the recommendation results varied as time changed. For the user 1500, song 212,724 ranked higher at time $t1$ than that at time $t2$. A similar phenomenon existed in the results of user 13,207, where song 190,396 ranked higher at time $t3$. The users' intrinsic preferences remain the same as time changed, therefore, this phenomenon was caused by the different user dynamic preferences at different times. In other words, DTNMR predicted that the user 1500 was more interested in song 212,724 when $t1$ than $t2$ since the dynamic preferences generated by LSTM changed. As a conclusion, DTNMR could automatically adapt to new users and learns new features when they listened to more songs, and LSTM could learn different dynamic preferences of the same user at different times.

　　To recommend songs to users without listening histories, DTNMR asked the users to select their interested genres or artists, and recommended relevant and popular songs based on their selection.

## 5. Conclusions

In this paper, we proposeda Deep Temporal Neural Music Recommendation model (DTNMR) to recommend songs for users. The recommendation results of DTNMR are based on music characteristics and users' tempral preferences. The music metadata are encoded as one-hot vectors and projected to low-dimensional space by Deep Neural Networks. The outputs of the DNN are the characteristics of the music. The users' temporal preferences are learned from their listening histories based on LSTM networks. With the help of user behavior type, DTNMR can evaluate the different importance of user listening histories. DTNMR alleviates the cold start problem in the item side and discovers a new users' preferences immediately after they listen to music. Since DTNMR recommends songs without any extra information, it can be deployed to all music platforms. The experiment results show that the DTNMR is the best among the seven baseline methods.

Since the description of users' intrinsic preferences in DTNMR is relatively simple, we will study learning users' intrinsic preferences with more complex models such as convolutional neural networks. Considering the DTNMR model cannot handle newly appeared artists or composers, we will study representing these new feature values based on existing similar songs. In addition, we will also study how to rate between music characteristics and user preferences more effectively. Some other distance functions may be more successful choices.

**Author Contributions:** H.-T.Z. and J.-Y.C. conceived and designed the experiments; J.-Y.C. and N.L. performed the experiments; H.-T.Z. and J.-Y.C. analyzed the data; C.-Z.Z. contributed reagents/materials/analysis tools; H.-T.Z. and J.-Y.C. wrote the paper; A.K.S. and Y.J. proofread the paper.

**Funding:** This research is supported by National Natural Science Foundation of China (Grant No. 61773229), Basic Scientific Research Program of Shenzhen City (Grant No. JCYJ20160331184440545), and Overseas Cooperation Research Fund of Graduate School at Shenzhen, Tsinghua University (Grant No. HW2018002).

**Conflicts of Interest:** The authors declare no conflict of interest.

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
