# Peer review of "A Deep Temporal Neural Music Recommendation Model Utilizing Music and User Metadata"

_applsci, doi:10.3390/app9040703_

Round 1

Reviewer 1 Report

This paper proposes a Deep Neural Model for Music Recommendation based on the music semantic information and the users’ temporal preferences.

This paper develops an important topic; the overall process is interesting. Nonetheless, there are some points that need to be clarified by the authors.

The authors emphasize on the fact that their model can handle the cold start problem, however, the authors need that new users listen to some songs, so they can infer recommendations, this is exactly the cold start problem. Indeed, as a part of the proposed model, the authors use Recurrent Neural Network to extract user’s dynamic preference from the user’s listening histories, but, what if there are no user’s listening histories?

Besides, the four components in the DNMR model make use of the songs listened to by the user whether at the short or the long term which means that the model cannot handle situations where there is no songs history

In Section 3.2, the authors indicate that they encode the field values as one-hot vectors separately. The concatenation result of all the fields is the meta vector of the corresponding song. The authors should indicate the function they use to aggregate the one-hot vectors of the field values?

A minor English revision is needed.

Author Response

The response to the reviewer's comments is in the uploaded PDF.

Best regards.

Jin-Yuan Chen

Reviewer 2 Report

Results must be cearly presented. The paper needs better state of the art methods comparison.

Author Response

(The authors gave the same response as above.)

Reviewer 3 Report

The authors present a music recommender in which they have combined meta-data and user preferences to solve the cold start problem. The combination provides moderate improvements over the system that only uses user preferences.

First of all, the title of the paper should be qualified since it is too generic. As it is now, it does not give enough information about the paper since there are many papers presenting deep-learning based recommender systems.

The related work and baselines should also consider content-based methods. In a way, using meta-data from the items is a variant of a content-based method and indeed a way to alleviate the cold-start problem. Surprinsingly, only collaborative filtering methods are described.

In the description of the paper, the authors call the predicted items from the meta-data semantic information which is quite misleading. Figure 1 should be improved to represent the fusion mechanism in a more detailed way. Regarding section 3.2, a more detailed description of the input vectors should be provided. For example, how sparse is the vector? How many artists are considered and how many composers? How do the authors cope with newly introduced artists or composers? The rationale for the configuration of the DNN should be provided (number of layers, type of activation function, etc.). Ideally, the hyper-parameters should be decided using a cross-validation set.

In eq. 2. the tildes should be wider to cover the two letters (UD, UD_t).

In table 2, the description of ListenCnt y SongCnt should be rephrased.

In eqs 5 and 6 the notation of i^+ is unfortunate because the way to represent time is not standard.

The last sentence of section 3.4 should be further explained.

In the experimental section further explanations and experiments are needed to assess the significance of the contribution:

- the authors claim that their system is capable of adapting to the evolution of the user preferences (the dynamics) due to the user preference module with the long and short-term components. However, this is never empirically demonstrated. In particular, in l163, the use of the so referred as to 'certain point' should be detailed.

- the authors use the data from a challenge (WSDM 2018) which is excellent to allow comparisons but the baselines do not come from this competition as they should. Moreover, details about the implementations used for the baselines should be provided (are they available from their authors, for example?).

- in section 4.2, the meaning of 'relevant fast' is not clear.

- in table 3, indicate that the figures are represented as a %.

- the sentence in l210-220 should be qualified since the improvements in performance are not significant (the confidence intervals are overlapped). Therefore it is very risky to say that the net 'understands' the different importance of the user learning histories. The conclusion section should also be modified to acknowledge this.

The paper is mostly well-written but I recommend a thorough revision of the verbs conjugations. Some of the typos are listed below.

Minor comments and typos:

- l50: extract--> extracted

- l65: needs --> need

-l91: relies-->rely

-l91: contens--> contents

-l92: neglects-->neglect

-l107: with --> to

-l110: etc.. : there is an extra dot.

l116: At last--> At the end

-l133: extract --> extracts

-l145: trow--> through

-l232:with-->to

Author Response

(The authors gave the same response as above.)

Round 2

Reviewer 3 Report

The authors have improved the paper but some of the request made previously are not properly addressed.

The State of the Art should mention results from content-based methods others than those based on metadata.

Most of my concerns have to do with the treatment of the time variable, both from the point of view of the proper description of the methods and the experiments:

Starting from the qualification of the tittle, I still do not see the main contribution of the paper reflected in the modified version. In p2, the authors list as the first contribution the modelling of the "users' temporal preferences".

In section 3, the algorithmia should be described mathematically in much more detail. Some improvements have been made in this direction but still there are variables that are undefined or with a vague verbal definition that does not allow for a full understanding. Moreover, the authors should explain the use of the tildes and the primes: for example, in eq 1 and eq 5. In eq 2 explain f and g, is the former related with the f in eqs 3 and 4? In eq 3, please, use \widetilde. Below eq 5 consider a better formulation for s_{u_1}, etc. since it could be interpreted as the song for user 1 instead of the song 1 for user u. In section 3.3.2, the authors need to formalize the concept of 'listening stories' and provide a proper mathematical formulation for this section. In section 3.4, eq 6 appears as a line of code but not as a proper equation, is it a simple linear combination? And again, in eqs 7 and 8, the use of s_{+} is confusing: is it the same t and u than the one in P (do you mean P(u,t, s(u,t))?. The answer (A5) is not satisfactory in relation to the hyper-parameters. A better explanation should be provided: hyper-parameters are usually set using separated development sets.

In section 4, the task from WSDM should be clearly described and the differences from the task that the authors solve should be clarified. Is it possible to directly compare the figures of merit from the challenge? Please, explain how does the goal of predicting 'the chances of a user listening to a song repetitively after the first observable listening event within a time window' relate to the goal of y in eq 6. Moreover, the role of the LSTM should be empirically determined for several t to support the conclusions.

Also, there are a number of typos in the new additions to the paper. Please, check them carefully. For example,

sec 3.2, l130: 'neural' should be 'neural network'

sec 3.3, l135: take advantage should be takes advantage

Author Response

(The authors gave the same response as above.)

Round 3

Reviewer 3 Report

The authors have fulfilled most of my requests.

Only minor modifications are needed:

In eq 2, g(f(x)) is not necessary at all and it only adds confusion because there is no x whatsoever and the second part of the equality provides the information one needs to know to obtain l_i. Also this leaves f free again to be used in eq 3.

In table 3, the input of DTNMR should be the user listening stories.

Please, rephrase p7, l193. The use of the word 'missed' here is misleading.

Typos:

p7 l188: 'interested' should be 'interesting'

Author Response

(The authors gave the same response as above.)
